



# A Submesoscale Eddy Identification Dataset Derived from GOCI I Chlorophyll–a Data based on Deep Learning

Yan Wang[1], Jie Yang[1,2], Kai Wu[1], Meng Hou[1], Ge Chen[1,2]

[1]School of Marine Technology, Frontiers Science Center for Deep Ocean Multispheres and Earth System, Ocean University of China, Qingdao 266100, China
[2]Laboratory for Regional Oceanography and Numerical Modeling, Laoshan Laboratory, Qingdao 266100, China

*Correspondence to*: Jie Yang (yangjie2016@ouc.edu.cn)

**Abstract.**

This paper presents an observational dataset on submesoscale eddies, which obtains from high–resolution chlorophyll–a distribution images from GOCI I. We employed a combination of digital image processing, filtering, YOLOv7–X, and small object detection techniques, along with specific chlorophyll image enhancement processing, to extract information on submesoscale eddies, including their time, polarity, geographical coordinates of the eddy center, eddy radius, coordinates of the upper left and lower right corners of the prediction box, area of the eddy's inner ellipse, and confidence score, which covers eight daily periods between 00:00 and 08:00 (UTC) from April 1, 2011, to March 31, 2021. We identified a total of 19,136 anticyclonic eddies and 93,897 cyclonic eddies at a confidence threshold of 0.2. The mean radius of anticyclonic eddies is 24.44 km (range 2.5 km to 44.25 km), while that of cyclonic eddies is 12.34 km (range 1.75 km to 44 km). The unprecedented hourly resolution dataset on submesoscale eddies provides information on their distribution, morphology, and energy dissipation, making it a significant contribution to understanding marine environments and ecosystems, as well as improving climate model predictions. The dataset is available at https://doi.org/10.5281/zenodo.7694115 (Wang and Yang, 2023).

## 1 Introduction

Submesoscale eddies (SMEs) are one of the strong ageostrophic submesoscale processes in the ocean with horizontal scales ranging from several to tens of kilometers and vertical scales of ten to hundreds of meters, which are intermediate between the mesoscale and the microscale, and they typically exhibit a short lifespan ranging from hours to days (McWilliams, 2019; Durand et al., 2010; Thomas et al., 2008). SMEs are often energized by the strong mixing induced by ocean currents' instabilities, the convergence of fronts, or the influence of topographic features (Thomas, 2012; Taylor and Thompson, 2023). SMEs play a crucial role in material and energy exchange, influencing biochemical cycles, marine food webs, and climate change (Lévy et al., 2012, 2018; Wang et al., 2022b). Their significance has made them a major focus of research in oceanography.



Numerical simulations are the main method currently used for the systematic study of SMEs. Numerical simulations allow researchers to study SMEs in detail, by generating a large amount of data that can be analysed to understand their characteristics, formation, and evolution (Zhang et al., 2020; Cao et al., 2021; Dong et al., 2020; Marchesiello et al., 2011; Chrysagi et al., 2021). Nevertheless, numerical simulations often involve idealizations of many parameters in fluid mechanics, which may deviate from the complex and constantly changing nature of the ocean (Garabato et al., 2022).

Meanwhile, other analytical methods such as satellite observations, in–situ measurements and laboratory experiments are still significantly insufficient for studying SMEs. There are two fundamental challenges in studying submesoscale processes. On the one hand, submesoscale processes have very small spatial and temporal scales, making direct field observations challenging. Currently, the feasible field observation schemes are too expensive and sparse to form systematic results that cover the globe (such as dense submerged buoy arrays, ship–based towed CTD measurements, etc.). On the other hand, there

is still a lack of a clear definition of submesoscale processes in terms of dynamics, and there is some controversy. It seems that submesoscale processes at least include frontal instability processes at the edges of mesoscale eddies, inertial gravity waves falling into submesoscale spatiotemporal scales, Vortex Rossby Wave on mesoscale eddies, and SMEs, etc(Zhang and Qiu, 2018).

The observation of SMEs has not ceased, and SAR images are being used to identify "black" and "white" eddies (Dokken

and Wahl, 1996; Fu and Ferrari, 2008; Xu et al., 2015; Ji et al., 2021; Hamze-Ziabari et al., 2022). Additionally, as early as 1980, oceanic submesoscale processes driving phytoplankton patchiness movement were observed (Gower et al., 1980). Observations of SMEs are carried out by manual labelling, algorithmic identification and machine learning methods (Park et al., 2012; Ni et al., 2021; Xia et al., 2022). Compared to the method of SAR images, it can only provide physical information about the ocean surface and not information about the biological or chemical processes within the eddies. In contrast, using

chlorophyll to identify eddies can provide information about the composition and activity of the biological communities within the eddies. Furthermore, the method of SAR images typically requires additional data processing and algorithms to accurately identify SMEs, which could make the identification process complex and time–consuming.

We used a combination of digital image processing, filtering, artificial intelligence, and small object detection techniques to identify a large number of SMEs from high–resolution chlorophyll fields and calculated their relevant characteristic

information to form a SMEs dataset. The paper is organized as follows: in Section 2.1, we provide a detailed description of the chlorophyll data used in the study. Next, in Section 2.2.1, we describe the methodology used to highlight SMEs in chlorophyll images. This is followed by Sections 2.2.2 and 2.2.3, where we elaborate on the machine learning recognition process. Finally, in Sections 3, 4, and 5, we present the results of our study, provide information on the acquisition of the dataset, and summarize the whole research.



## 2 Data and Methods

### 2.1 Chlorophyll–a Data

The chlorophyll–a (CHL) data used in this study were obtained by applying the OCI empirical algorithm to Level-2 data acquired by the Geostationary Ocean Color Imager I (GOCI) aboard the Oceanography and Meteorology Satellite (COMS) (https://doi.org/10.5067/COMS/GOCI/L2/OC/2014)(Ryu et al., 2012; Hu et al., 2012). These data have a spatial resolution of 500 meters and a temporal resolution of one hour, with measurements taken within an area of 2500 km × 2500 km (Center: 130° E, 36° N) and a 20–minute window between 0 UTC and 8 UTC from 1 April 2011 to 31 March 2021. The array size of the data is 5685 in the meridional direction and 5567 in the zonal direction. One unique feature of the GOCI is its geostationary orbit, which allows it to continuously observe the same region of the Earth without moving relative to the ground. This makes it particularly useful for monitoring dynamic ocean phenomena such as coastal currents, and ocean color. In Fig. 1, we show the coverage area of the GOCI.

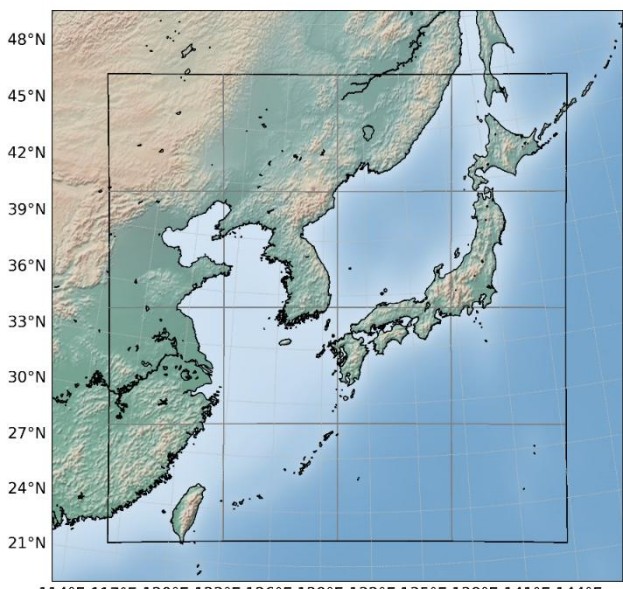

**Figure 1: The coverage area of Geostationary Ocean Color Imager I. The coverage area consists of 4 × 4 slots that overlap with each other (Lambert Azimuthal Equal Area Projection).**

### 2.2 Identification Method

### 2.2.1 Chlorophyll Image Enhancement

Firstly, we presented the flowchart of the CHL image enhancement technique in Fig. 2.



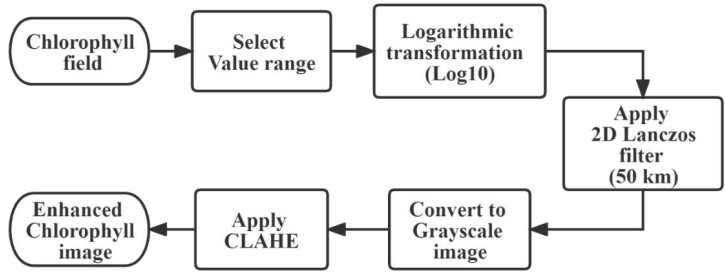

**Figure 2: The flowchart of the CHL image enhancement. We have selected a data range of 0–20 (mg/m3) based on the CHL value range of the satellite coverage area. Before applying CLAHE, we will assign a value of 0, i.e., black display, to invalid regions such as those occluded by clouds.**

The difference in CHL concentration between coastal and oceanic regions by several orders of magnitude makes it easier to directly manually interpret SMEs in coastal areas, while they are almost indistinguishable in regions with low CHL concentration. It is often necessary to apply a logarithmic transformation to CHL data when plotting CHL fields. This is done in order to avoid the problem of colour stacking displays that can occur when there are large differences in CHL concentration. By taking the logarithm of the data, the range of values is compressed and it becomes easier to distinguish between areas with different CHL concentrations. This transformation can help to create clearer and more informative CHL maps. However, using only logarithmic transformation is not enough, as shown in Fig. 3a. Large-scale circulation, mesoscale eddies, waves, and other processes at larger scales mask the CHL variability caused by submesoscale processes. We used a 2D Lanczos filter with a half-power cut–off wavelength of 50 km, consistent with the sea surface height field (refer to Fig. 3b)(Pegliasco et al., 2022).

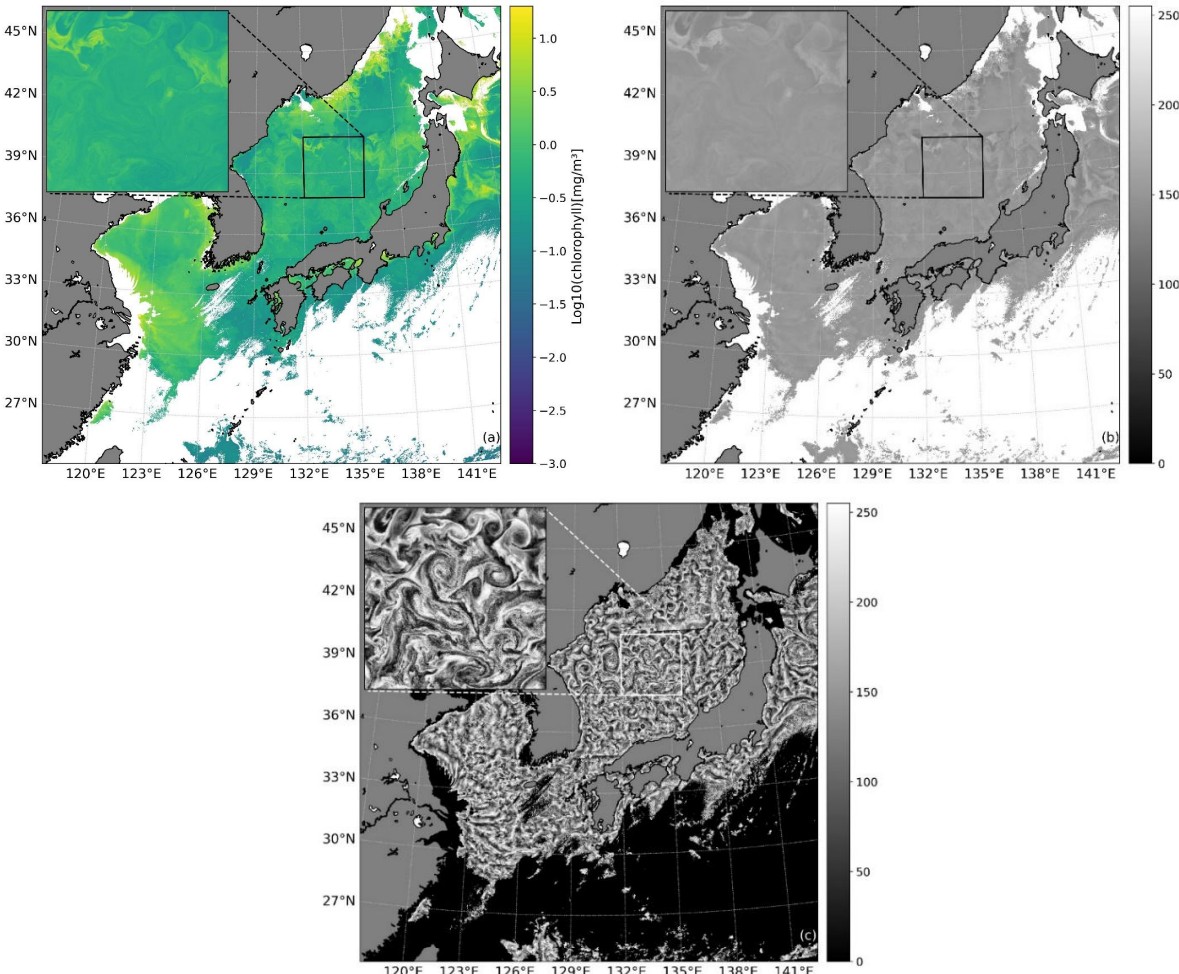

**Figure 3: The Comparison of different CHL image enhancement methods (April 5th, 2011 at 3 AM UTC). (a): The CHL values were selected in the range of 0-20 mg/m³ and then log10 transformation was applied to them. (b): Apply a 2D Lanczos filter with a half power cut–off wavelength of 50 km to image (a) and convert it to a grayscale image with a range of 0-255 for visualization. (c): The final effect of enhancing the entire CHL image by applying CLAHE to image (b). For each image, the top-left subfigure displays a 3x magnified view of the boxed region in the image, allowing for a clear visualization of the effect of each step in the image enhancement process on the high-resolution image.**

We tested the half–power cut–off wavelength of the filter and found that if the wavelength is too long, it will mask the spiral structure inside mesoscale eddies, making it difficult to distinguish SMEs and their polarity. On the other hand, if the wavelength is too short, it will generate more discontinuous vortex filaments and make it difficult to identify relatively closed SMEs (refer to Fig. 4).

**Figure 4: Comparison of the CHL image enhancement results using different filter cut–off wavelengths (April 5th, 2011 at 3 AM UTC). (a), (b), and (c) show the results obtained by selecting a half-power cut–off filter wavelength of 1 km, 5 km, and 200 km, respectively. Because the CLAHE technique can increase brightness and contrast, it reduces the contrast differences when the filter selects different longer wavelengths, as shown in Figure 3 and Figure 4 (c).**

However, this approach still cannot completely solve the problem of low gradient CHL mapping. We adopted a contrast limited adaptive histogram equalization (CLAHE) image enhancement technique to highlight the SMEs with the same display effect in the entire image (refer to Fig. 3c). Adaptive histogram equalization (AHE) is a widely used technique for image contrast enhancement, which calculates the image's histogram and applies a non-linear transformation to stretch the intensity values. But AHE can lead to excessive amplification of noise in relatively uniform areas of the image. Contrast-limited adaptive histogram equalization (CLAHE) is a modification of AHE that helps avoid this problem by limiting the amplification of the contrast to a certain predefined value (Zuiderveld, 1994; Vidhya and Ramesh, 2017). This approach



involves dividing the image into small regions, called tiles, and then applying AHE to each tile individually. The general histogram equalization formula is the following Eq. (1):

$$h(v) = \text{round}\left(\frac{cdf(v) - cdf_{min}}{(M \times N) - cdf_{min}} \times (L-1)\right) \tag{1}$$

Where $v$ represents the intensity of any pixel in the image, $h$ represents the histogram equalization function, $cdf$ is the cumulative distribution function of the image pixel intensities, $cdf_{min}$ is the minimum non-zero value of the cumulative distribution function, $M$ is the width and $N$ is the height of the image, and $L$ is the number of gray levels used (in most cases, 256).

Considering the horizontal scale of SMEs, a sliding window size of 100x100 was chosen when applying adaptive histogram equalization with contrast limiting. Moreover, to improve visualization and reduce the computational burden for machine learning, we processed the CHL data into a grayscale image.

### 2.2.2 Train Set Establishment

After thorough preprocessing of the CHL field, we are able to manually annotate the locations of SMEs relatively easily by using rectangular boxes in the enhanced CHL images. The discrimination between cyclones and anticyclones was based on the rotation direction of the eddy-modulated CHL spiral curves from the outside to the inside. This direction is consistent with the rotation direction of the two types in the Northern Hemisphere, where cyclones rotate counterclockwise and anticyclones rotate clockwise (Chelton et al., 2011; Zhang and Qiu, 2020; Wang et al., 2023). Due to the high image resolution and the large number of eddies present, it is not currently feasible to annotate the entire image to include all cyclonic and anticyclonic eddies as well as non-eddy regions and conduct machine learning training using either contemporary hardware or manual labour. Therefore, we adopted a labelling strategy for our annotations, which categorized labels into three types: Cyclone eddies (CE), Anticyclone eddies (AE), and bounding boxes (BOX). Subsequently, we extracted the BOX from high-resolution images as actual training images for the network. A total of 513 BOXs were annotated, including 160 anticyclones and 500 cyclones. To enhance model robustness and increase training sample diversity, data augmentation such as adding salt–and–pepper noise, histogram equalization, random angle rotating images, and adding random Gaussian noise to images were employed, as shown in Fig. 5.



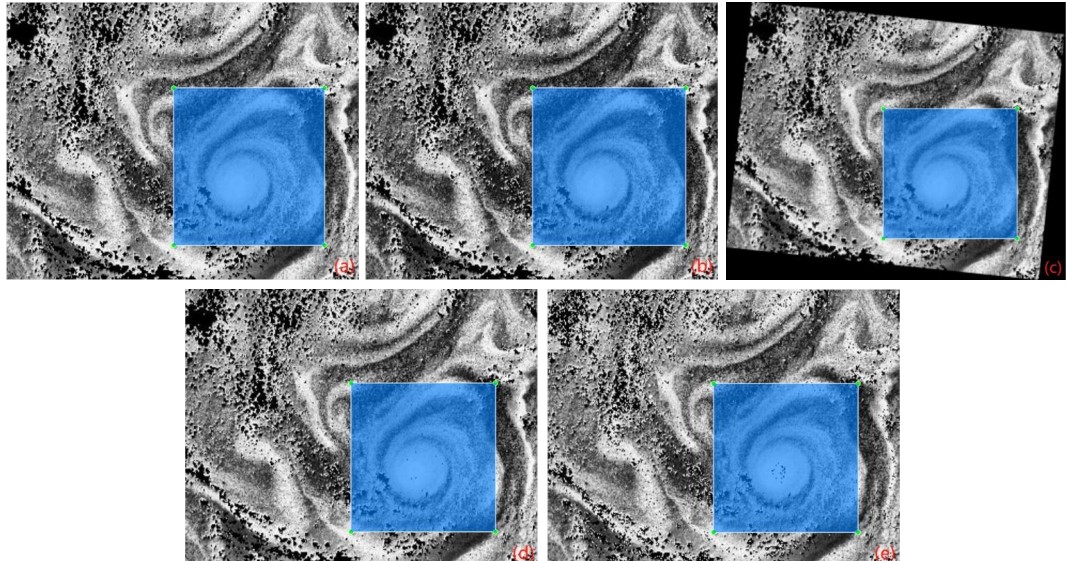

**Figure 5: Different approaches for the sample set augmentation. An example of a marked CE in a BOX, including the original**
**image (a), images modified using histogram equalization(b), random angle rotation(c), salt-and-pepper noise(d), and random**
**Gaussian noise(e). After rotation, the parallel marking box is the minimum bounding rectangle of the rotated marking box.**

### 2.2.3 Image Preprocessing and SMEs Identification

As is well known, target detection of high-resolution images or small targets is extremely challenging, where small targets
are defined as either having a relatively small size compared to the entire image or having a small difference in pixel value
compared to surrounding pixels. Obviously, SMEs fully comply with both definitions and they are ubiquitous and
intertwined with CHL fields. Therefore, we developed an image preprocessing method for identifying SMEs, which includes
an image cropping method based on the eddy radius and the conversion between the image and the geographical coordinate
system. The cropped image resolution is 640 * 640, and the overlap percent is calculated based on the diameter of the
submesoscale eddy, following Eq. (2):

$OP = D/(SR * PS),$                                                               (2)

Where $OP$ is the overlap percent, $D$ is the maximum diameter of SMEs (100 km), $SR$ is the spatial resolution (0.5 km), and
$PS$ is the size of cropped images (640). Therefore, it can be calculated that an original image with dimensions of 5685 * 5567
can be divided into 12 * 12 small images through cropping, and each cropped image will have its corresponding row and
column number in the original image. We have set a requirement that the effective CHL data rate in each cropped image
should not be less than 5%. We calculate the point geographic coordinates of the cropped image by the row and column
numbers of the cropped image and the transformation relationship between the image coordinate system and the geographic
coordinate system of the original image. If $(x, y)$ is an image coordinate point in the cropped image, then its geographic
coordinate ($lon$, $lat$) can be calculated as follows in Eq. (3):



$$lon, lat = f\left((x + \text{col} * PS\,(1 - OP)), (y + \text{row} * PS\,(1 - OP))\right), \tag{3}$$

Where the function $f$ describes the correspondence between the original image coordinates and the geographic coordinates, and col and row represent the column and row number of their corresponding cropped images in the original image, respectively. The flowchart of the overall process of identifying SMEs and generating datasets using enhanced chlorophyll images is shown in Fig. 6.

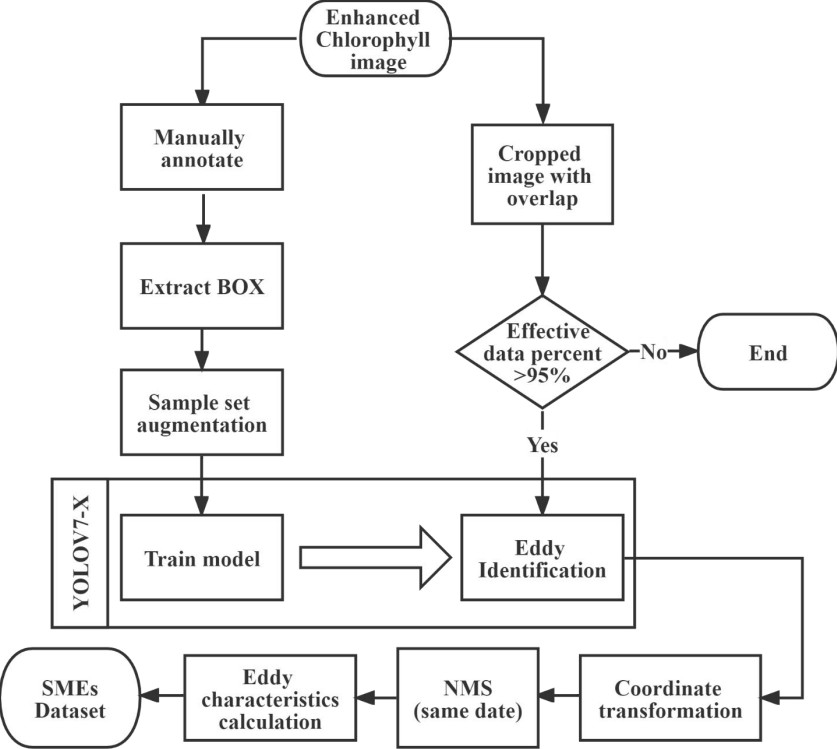

**Figure 6: The flowchart of the overall process of identifying SMEs and generating SMEs datasets using enhanced chlorophyll images.**

We used the YOLOv7–X as the model for the identification of SMEs, which balances speed and accuracy perfectly (Wang et al., 2022a). YOLOv7–X was obtained by performing stack scaling on the neck and using the proposed compound scaling method to scale up the depth and width of the entire model, based on YOLOv7. The structure of YOLOv7–X mainly consists

of three parts: backbone feature extraction network part, strengthen feature extraction network and YOLO Head. In order to accelerate model convergence and reduce memory consumption, the Adam optimizer is selected to learn the parameters of all models automatically. The loss function of our model inherits the loss function of the YOLO series, which mainly includes shape loss, confidence loss, and classification loss of the predicted box. The total loss function of object detection is defined by the following Eq. (4):

$$\text{Loss}_{total} = \text{Loss}_{shape} + \text{Loss}_{confidence} + \text{Loss}_{class}, \tag{4}$$



Where $\text{Loss}_{shape}$, $\text{Loss}_{confidence}$, $\text{Loss}_{class}$ denote the shape loss, confidence loss, and classification loss of the predicted anchor box, respectively, the confidence is a signal to judge whether the anchor box contains objects. Their basic components are binary cross-entropy loss and mean squared error loss (Redmon and Farhadi, 2018; Bochkovskiy et al., 2020; Ge et al., 2021).

Furthermore, to avoid repeated identification of eddies in the overlapping regions of the cropped images, a non-maximum suppression technique was utilized to merge them. Since many eddies are formed from the same unstable currents and often overlap, we set the intersection-over-union (*IoU*) threshold for non-maximum suppression to 20%. The *IoU* is the overlap ratio between the detected box (DT) and the corresponding ground truth box (GT). The *IoU* can be calculated by the following Eq. (5):

$$IoU = \frac{S_{DT} \cap S_{GT}}{S_{DT} \cup S_{GT}},\tag{5}$$

where $S$ represents the pixel areas of the anchor box, $S_{DT} \cap S_{GT}$ is the intersection area of *DT* and *GT*, and $S_{DT} \cup S_{GT}$ denotes their union area.

In addition, in order to avoid detecting incomplete eddies at the image edges, we removed the identification results within 5 pixels of the edges. We disabled the mirror transformation for image enhancement within the model and applied non-

maximum suppression for different categories of eddies to prevent the model from detecting the same eddy as having different categories.

**2.2.4 Cloud Cover in the Identification**

The results of SMEs identification obtained from the watercolor remote sensing images cannot represent the actual distribution of SMEs in the region. The primary issue is the obscuring of the ocean color remote sensing signals by cloud

cover. The coverage of clouds above the region is the primary obstacle that affects the identification of eddies using this method. Cloud coverage varies across different regions, different months of the year, and different times of the day. Therefore, we calculated the cloud occlusion probability (*cop*) for each *grid* at eight hours per *month* using invalid CHL data, as follows Eq. (6):

$$cop(time, grid) = \frac{\sum mask(time, grid)}{fn(time)},\tag{6}$$

Where $mask(time, grid)$ is a bool daily grid array (5685, 5567) of whether the data corresponding to hour and month are masked, and $fn(time)$ is the total number of the CHL files at the corresponding to hour and month. Therefore, by using *cop*, we can roughly calculate the number of detected eddies without cloud cover, as follows Eq. (7):

$$TN = \frac{EN}{1 - cop}\tag{7}$$

Where *TN* is the number of eddies detected after removing cloud cover, and *EN* is the actual number of detected eddies.



## 3 Result

### 3.1 Identification Results of SMEs

We obtained a total of 29,158 files from April 1, 2011, to March 31, 2021, amounting to approximately 7.3 terabytes of data (some files were missing due to a high rate of invalid data). The chlorophyll data were extracted and used for image enhancement, resulting in a corresponding number of images. Ultimately, we obtained a total of 544,760 cropped images for the identification of SMEs. A total of 19,136 anticyclonic eddies and 93,897 cyclonic eddies were identified at a confidence threshold of 0.2. As shown in Fig. 7, it can be seen that our method can effectively identify SMEs from the chlorophyll field, and the chlorophyll spirals traced by the SMEs indicate their position and size. In the AEs, the direction of rotation of the chlorophyll spirals from the outside to the inside is clockwise, while in the CEs it is the opposite. The higher the confidence of the identification results, the more the chlorophyll spirals resemble Fibonacci spirals. There are both individual SMEs, as well as horn-shaped eddies with different types and S-shaped eddies with the same type. From (c) and (d), it can be seen that although the two have different cloud coverage, the energy of the SMEs dissipates within just two hours, making it impossible to trace them in the chlorophyll field. The apparent horizontal dividing lines in (b) and (f) are traces of the joining of different slots. Using the CLAHE technique, these subtle stitching marks became visible. In fact, there are several minutes of measurement interval between different slots, which results in differences in chlorophyll values between overlapping slots. (e) and (f) demonstrate that the cropped images with a 100 km overlap can effectively prevent missed detections at the edges of the images, and the eddies recognized in the overlapping area are different, but they can be removed through non-maximum suppression.

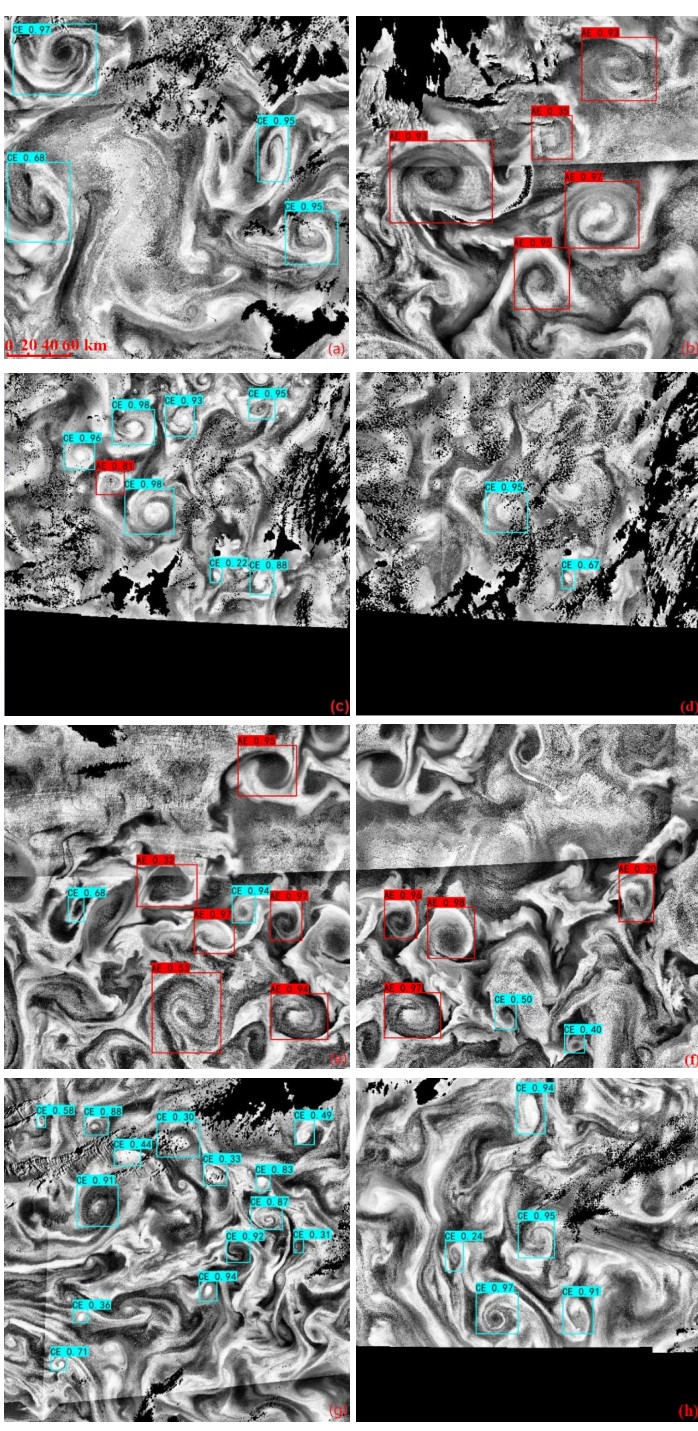

**Figure 7: Image identification results of SMEs. The blue box represents CE, the red box represents AE, and the number in the**
225 **upper left corner of each identification box is the confidence score; (c) and (d) show the identification results of the same location**
**at different times of the same day; (e) and (f) show the identification results of adjacent cropped images.**

## 3.2 Geographic and temporal distribution of SMEs

We counted the number of times each grid cell was covered by AEs or CEs and attempted to remove the high correlation between the spatiotemporal distribution of SMEs and a *cop* by the method of 2.2.4. From Fig. 8, it can be seen that AEs are mainly distributed in the Sea of Japan, which is a convergence zone of warm and cold currents, with the Kuroshio current passing through this area. On the other hand, CEs are more evenly distributed, with relatively more of them near offshore currents.

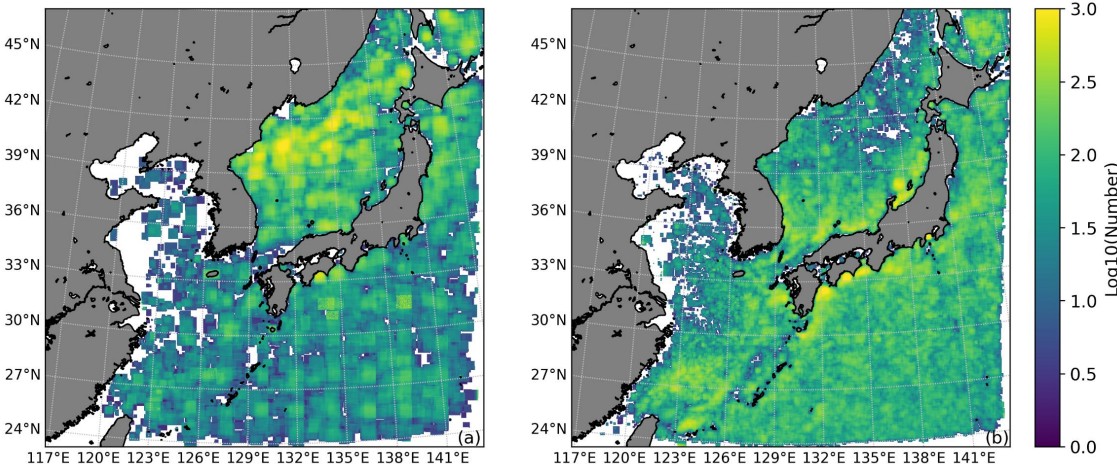

**Figure 8: Geographical distribution map of SMEs identified. The left figure is for anticyclones (a), and the right figure is for cyclones (b). The grids where eddies were identified were summed up by time and month after removing the cloud cover factor. Due to the large differences in the number of eddy geographical distributions, a logarithmic transformation was used for plotting.**

As shown in Fig. 9, both AEs and CEs exhibit similar patterns of variation in quantity with respect to hour and season. When calculating the local time at the central longitude of 130°E in the region, the highest number of identified SMEs occurs at around 11:40 AM. From the seasonal variation, the number of both AEs and CEs peaked in April, with another peak appearing in autumn. The peaks in the number of identified eddies are concentrated in the times of the strongest variations in sea surface temperature, salinity, and wind conditions.

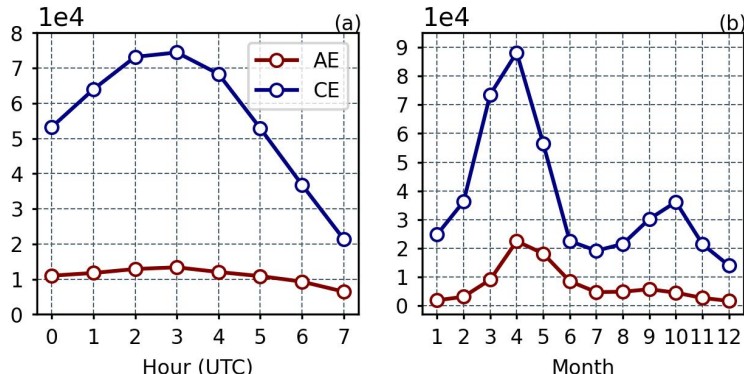

**Figure 9: Temporal variation in the number of identified submesoscale ocean eddies. (a): The figure shows the variation in the number of identified eddies over hours. (b): The figure shows the seasonal variation in the number of identified eddies.**

**3.3 SMEs Characteristic Statistics**

As shown in Fig. 10 (a) and (b), the diameter distribution of AEs is relatively uniform, while the radius of CEs is concentrated within 40 km, perhaps because the CHL field stirred by smaller-scale AEs is difficult to observe. In (c) and (d), observed AEs and CEs have the same confidence scores distribution and a majority of the detected eddies have high confidence scores. To better study SMEs, eddies with higher confidence scores can be selected for analysis. The observed

SMEs are non-geostrophic, and their diameter does not decrease with increasing latitude. We compare the estimated Rossby deformation radii. From (e) and (f), it can be seen that the diameters of SMEs at different latitudes can differ by about 30 km, and most CEs are smaller than the average Rossby deformation radius at the corresponding latitude.

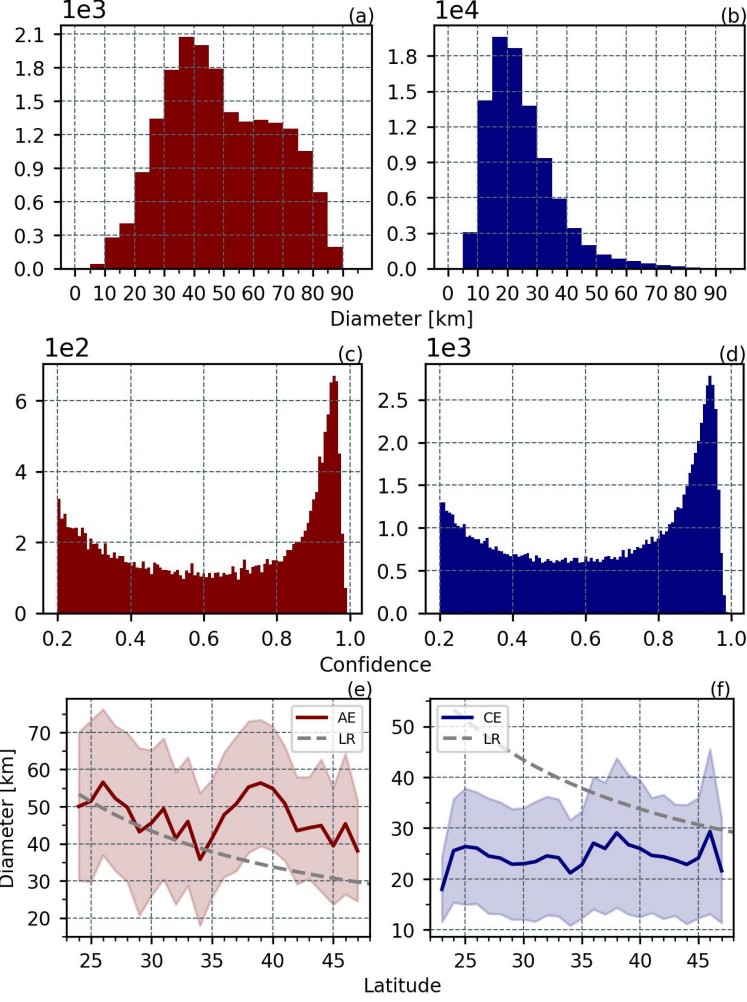



**Figure 10: (a) and (b) show the diameter distribution histograms of AE and CE, respectively; (c) and (d) show the confidence score distribution histograms of AE and CE, respectively; (e) and (f) show the variation of diameter with latitude for AE and CE and the standard deviation, respectively. where the gray dashed line represents the variation of the Rossby radius of deformation with latitude ($L_R = \frac{(g'D)^{1/2}}{f}$), Where $g'$ is the reduced gravitational acceleration, $D$ is the water depth, and $f$ is the Coriolis parameter.**

### 3.4 Performance of the Model for eddy identification

To evaluate the detection performance of the modified YOLOv7–X, some evaluation metrics were used: precision, recall, F1-score, average precision (AP), and mean average precision (mAP). The precision and recall are defined successively using equations (8) and (9):

$$\text{Precision} = \frac{TP}{TP+FP} \qquad (8)$$
$$\text{Recall} = \frac{TP}{TP+FN} \qquad (9)$$

where TP, FP, and FN denote the number of true positive, true negative, and false positive anchor boxes, respectively. In our experiment, the TP means the number of boxes whose $IoU$ is greater than 0.5 between the predicted and ground truth box. Besides, F1-score measures the comprehensive performance of the network, which can be calculated based on precision and recall.

$$\text{F1-score} = \frac{2\times\text{Precision}\times\text{Recall}}{\text{Precision}+\text{Recall}} \qquad (10)$$

The precision and recall of a specific category are used to draw curves in the 2-D coordinate system, and the area under the curve is AP of this category.

$$AP = \int_0^1 P(R)dR \qquad (11)$$

According to equation (11), mAP can be furnished, which represents the average of all categories of AP:

$$mAP = \frac{\sum_{i=1}^{n} AP_i}{n} \qquad (12)$$

The AP and mAP are commonly considered indicators of model quality. Generally speaking, the two indicators and model quality are positively correlated.

The various evaluation metrics in Table 1 demonstrate that the modified YOLOv7x model, trained on the samples processed and labelled with our method, has achieved outstanding performance. From the recall, fewer AEs were identified compared to CEs, although this could be due to a bias in the number of training sets for AEs and CEs.

**Table 1: Precision, Recall, F1–score and AP for different categories and Mean Average Precision at $IoU$=0.5.**

|    | Precision | Recall | F1-score | AP | mAP@0.5 |
|---|---|---|---|---|---|
| AE | 100.00% | 90.67% | 0.95 | 96.20% | 97.32% |
| CE | 97.80% | 96.52% | 0.97 | 98.44% | |

### 3.5 Validation and comparison of the identification results using Sentinel–3 chlorophyll image

Due to the differences in the GOCI and OCLI sensors, the blue-green spectral bands used for chlorophyll inversion are different, the calculation coefficients are different, and even the image resolutions are different. However, as shown in Fig. 11, this method has certain applicability. Comparing the identification results that differ by tens of minutes, the OCLI sensor

with a resolution of 300m has richer details and can identify S-shaped eddies that are not present in (c). However, since the model was trained using GOCI images, the confidence score of the eddy in (d) is lower.

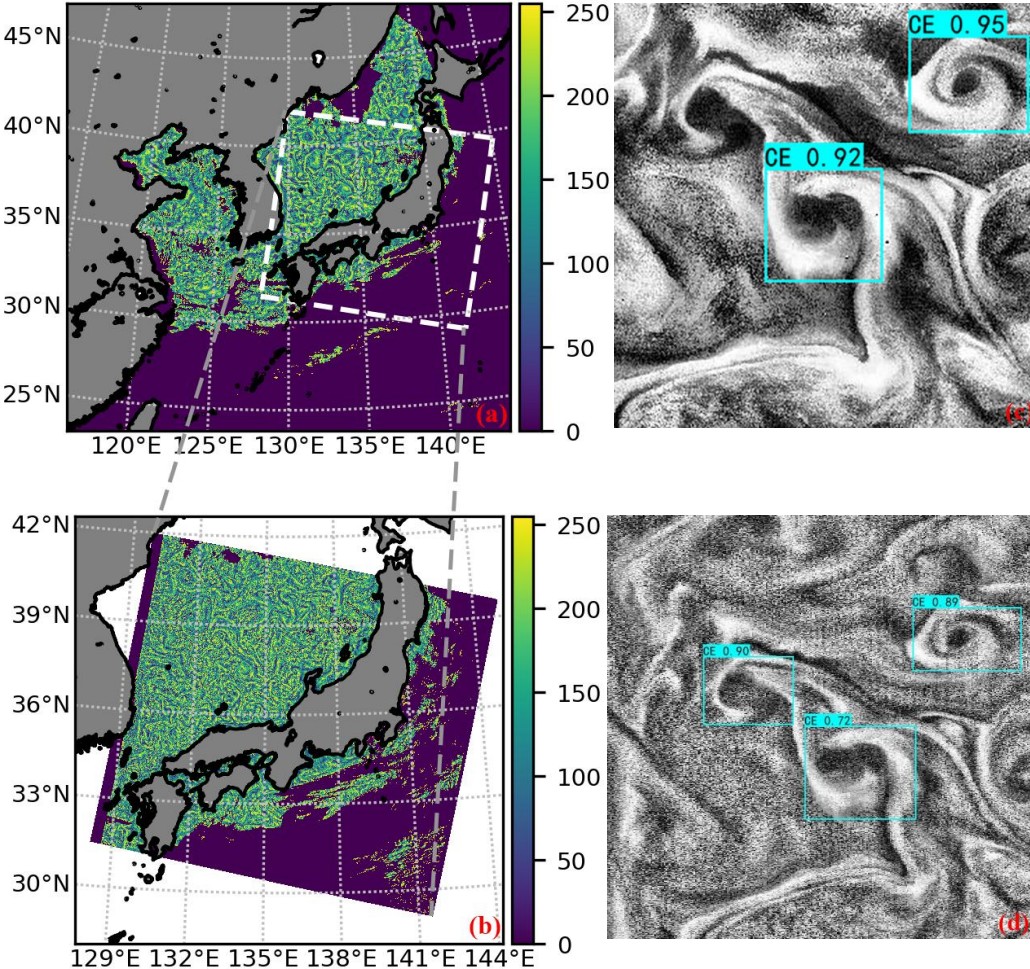

**Figure 11: A comparison of SMEs identified in chlorophyll images from the GOCI and Sentinel-3 OLCI sensors. (a) is the GOCI enhanced–chlorophyll image taken at 1:00 on May 7, 2019; (b) is the Sentinel-3B OLCI enhanced–chlorophyll image taken at a**

**similar time. (c) and (d) are the respective identification results of (a) and (b).**





**3.6 Validation and comparison of the identification results using the Mesoscale eddy dataset**

As it is well known, using altimetry can identify mesoscale eddies from sea level height data, but a daily mesoscale eddy dataset is identified by measuring different time orbits, which results in a reduction of spatial and temporal resolutions. We show the comparison between our identification results of SMEs and mesoscale eddies identified by altimetry on the same day in Fig. 12. Obviously, the altimeter identifies more eddies and can avoid the impact of cloud cover, but our method has higher spatial and temporal resolutions. The figure shows results that match well with the location and scale, as well as smaller-scale eddies that have been identified. It can be observed that there are smaller-scale eddies present in both the chlorophyll field inside and outside the mesoscale eddies. The eddies identified by the AI method are the mapping of their physical properties to the chlorophyll field.

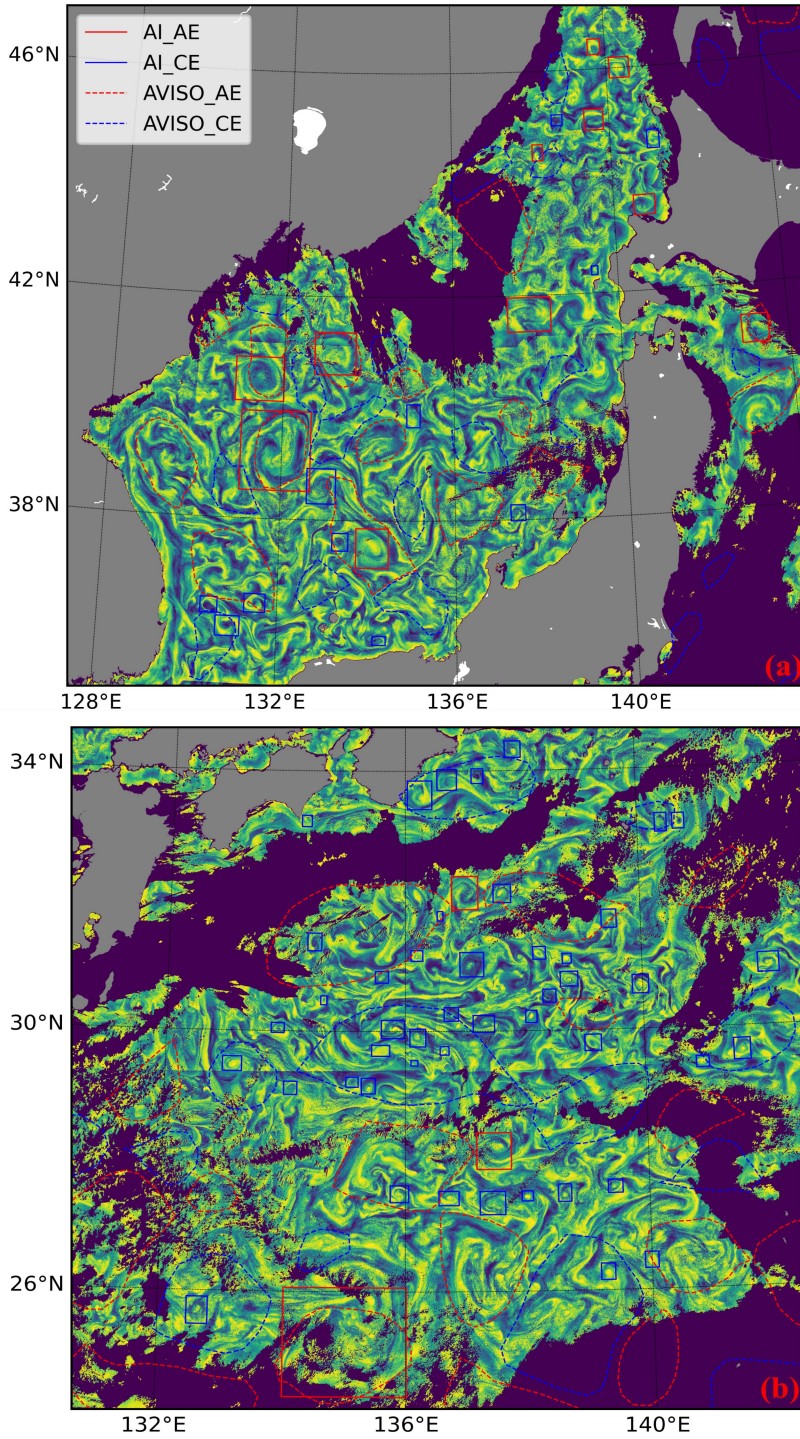

**Figure 12: the comparison between the AI vortex identification results and the AVISO vortex results on the same day with CHL-enhanced background. (a) and (b) are for April 13, 2011, and May 7, 2019, respectively.**



## 4 Datasets availability

The SMEs v1.0 dataset is saved in JSON format and can be accessed at https://doi.org/10.5281/zenodo.7694115 (Wang and
305 Yang, 2023). The dataset contains information about each identified eddy, including polarity, location, time, geographic
coordinates of the predicted box, radius of the inscribed circle, area of the inscribed ellipse, confidence score, and other
relevant information.

Other data used in this paper can be downloaded from the following websites:

GOCI I: https://oceandata.sci.gsfc.nasa.gov/directdataaccess/Level-2/GOCI (doi: 10.5067/COMS/GOCI/L2/OC/2014)

Sentinel-3B: https://oceancolor.gsfc.nasa.gov/data/10.5067/S3B/OLCI/L2/EFR/OC/2022 (doi: 10.5067/S3B/OLCI/L2/EFR/
OC/2022)

Mesoscale Eddy: https://www.aviso.altimetry.fr/en/data/products/value-added-products/global-mesoscale-eddy-trajectory-pr
oduct.html (doi: 10.24400/527896/a01-2022.005.220209)

## 5 Conclusion

Eddies can stir and maintain surface ocean chlorophyll, modulate temperature, mixing layer depth, and euphotic layer depth.
Therefore, eddies can be observed from the chlorophyll spirals structures at the sea surface, and with high spatiotemporal
resolution chlorophyll data from ocean color sensors, we suppressed large-scale ocean signals and increased chlorophyll
concentration gradients to highlight eddy-induced chlorophyll spirals with more significant contrast in different oceanic
environments. We modified YOLOv7–X for submesoscale eddy detection and achieved a map score of 97.32% for these
320 small targets. We identified a total of 19,136 anticyclonic eddies and 93,897 cyclonic eddies in ten eight-year periods at a
confidence threshold of 0.2. The number of cyclonic eddies was 4.9 times that of anticyclonic eddies, and the mean radius of
anticyclonic eddies was 24.44 km (range 2.5 km to 44.25 km), while that of cyclonic eddies was 12.34 km (range 1.75 km to
44 km). The mean radius of cyclonic eddies was half that of anticyclonic eddies, and the identified cyclonic eddies were
mainly concentrated in offshore flow regions, while anticyclonic eddies were mainly distributed in the Japan Sea. The
325 number of cyclonic and anticyclonic eddies followed the same pattern over time, increasing and then decreasing from around
9 am to 4 pm, with a peak around 12 pm. There were two peaks in the seasonal variation of both types of eddies, in spring
and autumn, both occurring when the mixed layer was relatively unstable. By comparing with chlorophyll products retrieved
from OLCI sensors using different bands at a resolution of 300 m, we found that the modified deep learning model had a
certain degree of universality. Compared with the mesoscale eddy dataset, the positions and sizes of the eddies identified by
the two methods were highly similar. Moreover, this method can detect SMEs, and the eddy-induced chlorophyll spirals
represent a direct mapping of eddy physical properties in the chlorophyll field, with high credibility. These research results
have important scientific significance for a deeper understanding of the role of SMEs in marine ecosystems and their impact
on the marine environment.



## 6 Author contributions

Conceptualization, Y.W.; methodology, Y.W.; validation, Y.W.; visualization, Y.W.; writing—original draft preparation, Y.W.; writing—review and editing, Y.W., J.Y., K.W., M.H. and G.C.; funding acquisition, J.Y and G.C. All authors have read and agreed to the published version of the manuscript.

## 7 Competing interests

The contact author has declared that none of the authors has any competing interests.

## 8 Disclaimer


Publisher's note: Copernicus Publications remains neutral with regard to jurisdictional claims in published maps and institutional affiliations.

## 9 Financial support

This research was jointly supported by the International Research Center of Big Data for Sustainable Development Goals
(No. CBAS2022GSP01), the National Natural Science Foundation of China (Grant No. 42030406 and No. 42276203), and the Ocean University of China (Grant No. 202251004).





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
