# Peer review of "A Submesoscale Eddy Identification Dataset Derived from GOCI I Chlorophyll–a Data based on Deep Learning"

_Earth System Science Data, 2023_

## Author Comment (AC1)

This manuscript describes a submesoscale eddy dataset derived from satellite ocean color products, which can be very useful for the studies of eddy dynamics and the ecosystem of oceanic environments. Overall, data and method to generate the dataset are well described, results are validated, and information for data access is complete. However, there are various grammar issues and fuzzy descriptions, which should be revised/corrected before publication. Specifically,

1. Line 9, "… an observational dataset on submesoscale eddies, which obtains from high–resolution chlorophyll–a …", clearly the grammar is not correct, should be something like "which was obtained from …"

**Response:** We feel great thanks for your professional review work on our article. I have changed it in line 9. "which was obtained from…"

2. L14, "which covers eight daily periods between 00:00 and 08:00 (UTC) from April 1, 2011, to March 31, 2021", this sentence belongs to a new sentence.

**Response:** Thanks for your comments. In deep learning, the range of confidence ranges from 0 to 1. I have changed it in line 14. "The dataset covers eight daily periods between 00:00 and 08:00 (UTC) from April 1, 2011 to March 31, 2021."

3. L15, "at a confidence threshold of 0.2". Need to state this 0.2 is high confidence or low confidence.

**Response:** Thanks for your comments. From the identification results, we find that the eddies with confidence greater than 0.2 are basically reliable and retain most of the eddies. I have changed it in line 15. "A total of 19,136 anticyclonic eddies and 93,897 cyclonic eddies were identified with a confidence minimum of 0.2.". Additionally, I added an explanation in line 210. "The higher the confidence of the identification results, the greater the reliability of identification results."

4. L40, "and there is some controversy" Need citations to support this statement.

**Response:** Thanks for your comments. I heard this from a lecture on submesoscale processes, and submesoscale dynamics are still developing. I'm sorry I didn't go into too much detail about submesoscale dynamics, so I decided to delete that sentence.

5. L42, ", etc(Zhang …"   It should be "etc.", and there should be a space before "(".   Please check the entire manuscript for similar issues.

**Response:** Thank you very much for your careful review. I have checked and corrected everything. (Line 43)

6. L50, "chlorophyll" here should be phytoplankton, as concentration of chlorophyll is a proxy for phytoplankton.

**Response:** Thanks for your comments. It is probably better to think about phytoplankton in conjunction with that rather than just say chlorophyll, which I have substituted. (Line 50)

7. **L54, "from high–resolution chlorophyll", note that "high-resolution" is subjective, and a resolution at 500 m is not "high-resolution" by many standards or measures.**

**Response:** Thanks for your careful comments. The high resolution here refers to chlorophyll retrieved by remote sensing technology. I have modified it in line 54. "high–resolution chlorophyll distribution images"

8. **L75, "Chlorophyll Image Enhancement" à "Enhancement of Chlorophyll Image". Make similar changes for 2.2.2.**

**Response:** Thanks for your comments. I have amended both to "Enhancement of Chlorophyll Image" and "Establishment of Train Set".

9. **L82, "directly manually ..", this is confusing.**

**Response:** Thanks for your comments. I have modified it for "artificial visual interpret". (Line 82)

10. **L92, "The Comparison of different …" should be "A comparison of different ..". Please also check similar issues at other places.**

**Response:** Thanks for your comments. I have revised all of them.

11. **L128, "where cyclones rotate counterclockwise and anticyclones rotate clockwise." This is common knowledge, no need to state here.**

**Response:** Thanks for your comments. Indeed, it is a common question, but many people in the computer field have asked me about how the training dataset was established to distinguish and classify different polarities of eddies, as well as the criteria used. I didn't delete it.

12. **L167, "We used the YOLOv7–X as the model", need citation for this model.**

**Response:** Thanks for your comments. I have a quote at the end of this sentence. The github address of the model can be found in this reference.

13. **L168, "YOLOv7–X was obtained by performing stack scaling on the neck and using …" This sentence is confusing, please rephrase.**

**Response:** Thanks for your comments. I restate it as "YOLOv7–X was obtained by increasing the number of layers and the number of features extracted per layer in the YOLOv7 model, aiming to amplify the model for improved performance in object detection tasks." (Line 167)

14. **L193, "the watercolor remote sensing images". Not such a thing of "watercolor remote sensing". It is either ocean color remote sensing, or "water color" remote sensing, but the latter is very rare.**

**Response:** Thanks for your comments. I have corrected it. (Line 192)

15. **L194, "images cannot represent the actual distribution of SMEs in the region." What does this mean?**

**Response:** Thanks for your comments. I'm sorry I made a mistake. What I mean is that the direct results of identification cannot be directly used to calculate the geographical distribution pattern. The influence of cloud occlusion needs to be removed. I made the following modifications. "The results of SMEs identification obtained from the ocean watercolor remote sensing images cannot represent the actual distribution pattern of SMEs in the region." (Line 192)

16. **L195, "The coverage of clouds above the region is the primary obstacle that affects the identification of eddies using this method." This is nearly identical to the previous sentence.**

**Response:** Thanks for your comments. I have merged two sentences into one. "The primary obstacle that affects the identification of eddies using this method is the obscuring of ocean color remote sensing signals by cloud cover, which varies across different regions, different months of the year, and different times of the day." (Line 193)

17. **L205, Result à Results.**

**Response:** Thanks for your comments. I have corrected it.

18. **L216, "the energy of the SMEs dissipates within just two hours, making it impossible to trace them in the chlorophyll field"   Why 'impossible'?**

**Response:** Thanks for your comments. I'm sorry, I'm being too absolute, but it's impossible for this model to recognize. I have changed "impossible" to "difficult".

19. **L220, "(e) and (f) demonstrate". This is not professional description, should be "Figs. (e) and (f) …"   Please also correct other similar places.**

**Response:** Thanks for your comments. I have corrected all references to subgraphs in the article.

20. **L237, please insert space between figure title and main text. Do so at other places.**

**Response:** Thanks for your comments. I have corrected everything.

21. **L258, "Performance of the Model for eddy identification", why "M" is capital? Style of headings or subheadings should be consistent.**

**Response:** Thanks for your comments. I have unified the style of all the headings and subheadings.

22. **L281, "Validation and comparison of the identification results using Sentinel–3 chlorophyll image". Why compare with results using Sentinal-3?**

> Note that the spatial resolution between GOCI-I and Sentinel-3 is similar, so similar results in eddy observation are expected.

**Response:** Thanks for your comments. This comparative experiment is of great significance. 1. Although the spatial resolutions of the two are similar, it can demonstrate that the differences in the measured central wavelengths and bandwidths of each spectral band will not affect the recognition performance of the model trained using the training set produced by GOCI, proving the model's certain universality. 2. Sentinel-3 provides global coverage with its dual satellites, which is important for future applications of this method in identifying submesoscale eddies at any location worldwide. 3. The chlorophyll algorithms used in the two instruments are also slightly different, further demonstrating their robustness.

**23.** L282, "Due to the differences in the GOCI and OCLI sensors, the blue-green spectral bands used for chlorophyll inversion are different, the calculation coefficients are different, and even the image resolutions are different." The reasoning is strange.

**Response:** Thanks for your comments. Indeed, from Figure 11, it is not apparent that these differences have a significant impact on the recognition results. However, these quantitative differences are mitigated by the unified process of chlorophyll image enhancement outlined in the article, making them less pronounced and enhancing the model's universality. For instance, when conducting recognition using images with different resolutions, parameters related to image segmentation and model recognition need to be adjusted. This experiment is beneficial for directly applying the methods and well-trained model from the paper to batch identification of submesoscale eddies in ocean color products with similar resolutions.

**24.** Paragraph below section 3.6. Many grammar issues, descriptions are confusing.

**Response:** Thanks for your comments. section 3.6. is modified as follows.

"As it is well known, altimetry is commonly used to identify mesoscale eddies through sea level height data. using altimetry can identify mesoscale eddies from sea level height data, but However, a daily global mesoscale eddy dataset is obtained by optimal interpolation is identified by measuring different time orbits, which results in a reduction of spatial and temporal resolutions. Therefore, We show the comparison between our identification results of SMEs and mesoscale eddies identified by altimetry on the same day in Fig. 12. Obviously, eddies identified by altimetry is morethe altimeter identifies more eddies and the method can avoid the impact of cloud cover, but our identification results of SMEsour method has higher spatial and temporal resolutions. Many of the eddies identified by both methods exhibit consistent spatial scales and locations. However, there are numerous submesoscale eddies that the altimeter fails to identify. These submesoscale eddies are found both within and outside the mesoscale eddies.The figure shows results that match well with the location and scale, as well as smaller-scale eddies that have been identified. It can

be observed that there are smaller-scale eddies present in both the chlorophyll field inside and outside the mesoscale eddies. The eddies identified by the AI method are the mapping of their physical properties to the chlorophyll field."

**25.** **L316, "from the chlorophyll spirals structures at the sea surface" Please check grammar.**

**Response:** Thanks for your comments. I have changed "at" to "of".

**26.** **L316, "… and with high spatiotemporal resolution chlorophyll data from ocean color sensors, we suppressed large-scale ocean signals and increased chlorophyll concentration gradients to highlight eddy-induced chlorophyll spirals with more significant contrast in different oceanic environments" Confusing sentence.**

**Response:** Thanks for your comments. I'm sorry that my explanation makes reading difficult. I have revised it as follows. "Therefore, eddies can be observed from the chlorophyll spirals structures of the sea surface. With high spatiotemporal resolution chlorophyll data from ocean color sensors, we suppressed large-scale ocean signals by filtering and highlight eddy-induced chlorophyll spirals by specific image enhancement."

**27.** **L320, "in ten eight-year periods" So, a total of 80 years? That is impossible.**

**Response:** Thanks for your comments. Sorry, I have modified it. "We identified a total of 19,136 anticyclonic eddies and 93,897 cyclonic eddies for eight times a day for a total of ten yearsin ten eight-year periods at a confidence threshold of 0.2."

**28.** **L330, "this method can detect SMEs, and the eddy-induced chlorophyll spirals represent a direct mapping of eddy physical properties in the chlorophyll field, with high credibility." Confusing sentence.**

**Response:** Thanks for your comments. I have revised the sentence as follows. "The method proposed in this paper successfully detects SMEs, and the presence of chlorophyll spirals induced by SMEs serves as a credible and direct representation of their physical properties within the chlorophyll field."

Finally, we appreciate for reviewer warm work earnestly and hope that the correction will meet with approval.

---

## Author Comment (AC2)

**This paper introduces an observational dataset of submesoscale eddies in the Northwest Pacific using deep learning techniques. While the approach and resulting product are novel, certain crucial results and discussions are missing. Specifically, this article exhibits significant language issues, including numerous grammar errors and unclear expression. I might consider accepting this article after these issues are truly resolved.**

**(1) Even though a precise definition of 'submesoscale eddy' is not yet established, the authors should provide a descriptive introduction to the fundamental characteristics (shape, size, structure, etc.). This is crucial for readers to comprehend the dataset. The authors' efforts in reviewing previous research are incomplete, as there is no mention of Munk's groundbreaking work in 2002.**

Response: Thanks for your comments. I read "Spirals on the Sea", and its three questions tried to answer initially explained some of the fundamental characteristics of spirals. The first paragraph of the introduction describes the spatio-temporal characteristics, structure, formation and role of submesoscale eddies (SMEs). I added the following sentence in line 25 "SMEs' spirals on the sea are considered a result of the cat's eye circulation associated with horizontal shear instability (Munk et al., 2000)". Currently, the introduction is enough as describing the detection of SMEs, despite many attempts in various articles to elucidate additional characteristics, such as the reason for the superior quality of cyclonic spirals over anticyclonic spirals, and this involves a lot of theories about the physical oceans.

**(2) Compared to logarithmic transformation, the CLAHE image enhancement technique can provide clearer information about spiral structures, but whether the enhanced signals are genuine and whether they might exaggerate the size and intensity of submesoscale eddies, these aspects need to be elucidated through some results.**

Response: Thanks for your comments. Another example is that CLAHE technology is also applied to medical images to identify lesions more clearly(Sonali et al., 2019). From a mathematical point of view, the CLAHE technique only performs the operation of increasing or decreasing the pixel value and does not create a spiral structure, let alone change the size of the original spiral or the density of the spiral. Additionally, the spirals have only become clearer, rendering them easier to identify by both manual observation and machine recognition from the zoomed-in Fig. 3 (a) and (c). And the main purpose of this dataset is to find out their location regardless of intensity.

**(3) L125. Prior to conducting large-scale identification, the utilization of manual annotation methods is required, undoubtedly introducing significant uncertainty. The authors need to demonstrate that the results of manual annotation are statistically reasonable. Figure 5 presents an eddy with a clear structure. The question arises regarding how eddies with less distinct structures are handled. This also touches on the issue of the definition of submesoscale eddies.**

**Response:** Thanks for your comments. Annotation work has been a major source of uncertainty affecting machine learning outcomes. In the AI identification of mesoscale eddies, people can use the results of the altimeter as the "real eddies" for labelling and model training. Further, the identification of mesoscale eddies by altimeter also requires a tolerance range to define the closed contours and a threshold to determine the size of the eddies. However, there are no conventional algorithms that can adjust physically meaningful parameters to provide "true" SMEs. Therefore, we usually analyze the results to see if our annotation set is missing some other kind of spiral structure such as not isolated, irregular, or more ambiguous. It is worth mentioning that the labelling work lasted for three months and was carried out by two independent people to prevent the wrong labelling of the direction of eddies or the eddies whose spiral structure was not clear. Finally, mAP@0.5 reached 97.32%. Additionally, if you still doubt our annotation set, I can upload it to Zenodo for people to view or would you like to provide what specific statistical analysis was done on the annotation set?

**(4) There have been some studies utilizing machine learning methods to detect mesoscale eddies in the ocean. The authors should introduce the related works and highlight the distinction between the submesoscale eddies identified here and mesoscale eddies. Is the difference merely in terms of size?**

**Response:** Thanks for your comments. I made a few additions to the introduction in lines 59-64 "Many studies have utilized machine learning methods to detect, track, and predict mesoscale eddies, owing to the abundance of reliable altimeter observations and the well-developed theory surrounding them(Duo et al., 2019; Choi and Kim, 2018; Franz et al., 2018; Ge et al., 2023; Huang et al., 2022). However, theoretical studies of SMEs lack sufficient observational information because the spatial and temporal resolution of the altimeter is not sufficient for observing them. Despite the availability of other high-resolution observational methods, submesoscale processes are obscured by a variety of large-scale ocean information". I want to emphasize that the significance of this paper is to provide a large number of eddies that cannot be observed by the altimeter for further SMEs research.

**(5) L210. 'at a confidence threshold of 0.2'. This is an exceptionally vital parameter, capable of greatly influencing the eventual product. The authors need to provide a clearer reason for the adoption of this value by means of sensitivity testing. This step is indispensable to eliminate artificial selection and ensure robustness.**

**Response:** Thanks for your comments. I have used this dataset to do some chlorophyll-related analysis and got some results similar to the simulations. I think the eddies above 0.2 confidence are sufficient for statistical analysis. There are some non-artificial interference methods to determine the value of confidence, such as adding confidence as a parameter to the loss function for model training to obtain higher mAP, but in industry applications, the actual effect of identification is more reliable than these parameters. The confidence of 0.2 was chosen because many eddies below 0.2 were wrong so I chose to keep eddies above the confidence of 0.2. It is preferable to cut some SMEs, but also to improve the reliability of the analytical results generated by the dataset.

Additionally, I can upload the full eddies data of the confidence from 0-1. Below Figure I are a few low confidence images.

[Figure]

**Figure I: Image identification results of SMEs.**

We also verify some of the results with different confidence thresholds. As shown in Figure II below, different confidence levels do not affect the conclusions.

[Figure]

**Figure II:** (a)(c) and (b)(d) show the diameter distribution histograms of AE and CE, respectively. (e) and (g): The figure shows the variation in the number of identified eddies over hours. (f) and (h): The figure shows the seasonal variation in the number of identified eddies. (a)(b)(e)(f) is the results with a confidence minimum of 0.5 and the confidence minimum of (c)(d)(g)(h) is 0.8.

**(6) L230. '…, with the Kuroshio current passing through this area'. Do you mean that the Kuroshio passes through the Sea of Japan?**

**Response:** Thanks for your comments. I'm sorry for the misunderstanding, but I meant that part of the Kuroshio flow into the Sea of Japan. I made changes in line 409 "It is evident that AEs are mainly distributed in the Sea of Japan along the convergence zone of warm and cold currents."

**(7) L245. Beyond location and size, is it possible to analyze the lifecycle of submesoscale eddies?**

**Response:** Thanks for your comments. It is entirely possible to track SMEs, but it is certainly more difficult than tracking mesoscale eddies, because the morphology and position of SMEs change faster over time, and a large number of observations are obscured by clouds. I think it's a challenging job.

**(8) Sections 3.5 and 3.6 do not show the validation of the detected eddies. These submesoscale eddies are derived from processed chlorophyll images. Can the authors utilize additional observational data to confirm the authenticity of these eddies, for example, high-resolution SST data or other flow observations?**

**Response:** Thanks for your comments. The data observed by remote sensing satellites are often used to prove the authenticity of simulation data, which is a kind of measured data. Additionally, the spatio-temporal resolution of the ocean current data generated by non-simulation is not sufficient to verify the SMEs of observation.

To evade potential contingency arising from single-sensor data, the observation of consistent Chlorophyll (CHL) distributions across different sensors is depicted in Section 3.5. Furthermore, Chapter 3.6 reveals the correlation between the altimeter and the chlorophyll field in terms of their shared eddies. Moreover, the high-resolution chlorophyll field can identify more SMEs.

We have searched the flow field data, but the measured ocean current data do not have a similar temporal-spatial resolution and many of them are interpolated data of reanalysis. As shown in figure III below, the coincidence between the two is not high due to the difference in temporal-spatial resolution, which is why the mesoscale eddies data is directly compared in Section 3.6.

("The GlobCurrent data repository now includes the surface geostrophic current, the Ekman current at the surface and at 15 m depth, and the combined geostrophic and Ekman currents. The data are interpolated and collocated to a common grid with a spatial resolution of 25 km and a temporal resolution of 1 day for the geostrophic current and three hours for the Ekman currents and the combined currents."
https://woc.oceandatalab.com/?from=globcurrent&date=1557230400000)

[Figure]

**Figure III: A comparison from the Sentinel-3 OLCI enhanced CHL image (left) and ocean current image (right) on May 7, 2019.**

Regarding the question of observational validation using high-resolution SST data, we believe that this is more likely to be a new question about the modulation of SST by SMEs. The VIIRS can provide the SST data with a spatial resolution of 750 m at nadir. However, it is not apparent for direct observation.

**(9) The color scheme of Figure 12 needs to be changed, as it doesn't clearly present the details.**

**Response:** Thanks for your comments. I changed the color of the cyclone box from blue to white as shown in figure V.

[Figure]

**Figure V: A comparison between the AI vortex identification results and the AVISO vortex results on the same day with CHL-enhanced background. (a) and (b) are for May 7, 2019, and April 13, 2011, respectively.**

**(10)  This dataset is regional in nature, focusing on submesoscale eddies in the Northwest Pacific Ocean. This point needs to be clarified in the title of the article, otherwise, readers might assume it's a global eddy dataset.**

**Response:** Thanks for your comments. The "GOCI I" in the title indicates the region and period of the dataset, and Figure 1 shows the coverage area of GOCI I. Our method for identifying SMEs is globally applicable.

**(11) For a dataset, especially results derived from observations, there are bound to be certain limitations. The authors need to engage in a discussion in this regard, providing readers with guidance and reminders when utilizing the dataset.**

**Response:** Thanks for your comments. I added some discussion on line 560. "However, there are some limitations in the use of this dataset. First, users should be mindful of the potential for underrepresentation or misidentification of certain features. Second, there is no clear physical definition to determine the boundary of the identified submesoscale eddies. Furthermore, the setting of the confidence threshold may delete a large number of real SMEs, to avoid retaining the identification of disputed eddies. Therefore, careful consideration must be given to the selection of the confidence threshold to satisfy the need of certain research. Nonetheless, the method proposed in this paper successfully detects SMEs, and the presence of chlorophyll spirals induced by SMEs serves as a credible and direct representation of their physical properties within the chlorophyll field. These research results have important scientific significance for a deeper understanding of the role of SMEs in marine ecosystems and their impact on the marine environment."

**(12) For the released product, an explanatory document needs to be added to clarify the meanings of various variables and provide instructions for processing the data.**

**Response:** Thanks for your comments. I wrote a "ReadMe.txt" file and uploaded it to Zenodo. The document is used to clarify the meanings of various variables of the dataset and provide an example of processing the data using Python code. You can choose to click the URL below to read (https://zenodo.org/record/8254335/files/ReadMe.txt?download=1).

The file contents are as follows:

The document is used to clarify the meanings of various variables of the dataset and provide an example of processing the data using Python code.

The name of each folder represents the UTC of the files inside.

| Variable name | Description | Units or Type |
|---|---|---|
| time | The time of obtained chlorophyll–a distribution image. | 'YYYYMMDD' |
| AE_sum | The number of anticyclonic | |
| CE_sum | The number of cyclonic | |
| predict | Prediction results in an image coordinate system derived from the deep learning model. | Array[n][7]* |
| eddy_type_AE0_CE1 | The type of eddy (0: anticyclonic; 1: cyclonic) | Array[n] |
| center_lon_lat | The longitude and latitude coordinates of the eddy center pixel. | Array[n][2] |
| box_min_lon_lat | The longitude and latitude coordinates of the pixel in the upper left corner of the rectangular box. | Array[n][2] |
| box_max_lon_lat | The longitude and latitude coordinates of the pixel in | Array[n][2] |

| | the bottom right corner of the rectangular box. | |
|---|---|---|
| inradius | The radius of the circle inside the rectangular box | Array[n](meter) |
| internal_ellipse_area | Area of the internal ellipse of the rectangular box | Array[n](m²) |
| confidence | Confidence of each eddy identification. Eddies with confidence levels below 0.2 were considered to be undesirable for data analysis. | Array[n] [0.2,1] |

*Array[n][7] represents a two-dimensional array of n rows and 7 columns.

   n is the sum of the number of cyclones and anticyclones.

You can perform eddy analysis by Python, or you can download other matching files such as chlorophyll, salinity, and temperature data for matching analysis.

The following example code plots the diameter distribution histograms of anticyclonic and cyclonic by the dataset.

```python
1.   import numpy as np
2.   import glob
3.   import pickle
4.   import json
5.   import matplotlib.pyplot as plt
6.   import matplotlib.ticker as ticker
7.   from tqdm import tqdm
8.
9.   geo_all = [np.array([]), np.array([])]
10.  file_pre = 'E:\\predict\\'   # The file path needs to be changed
11.
12.  for i in range(8):
13.      str_i = '0' + str(i) + '/'
14.
15.      for month in range(12):
16.          str_month = '0' + str(month + 1) if month < 9 else str(month + 1)
17.          print(i, ' ' + str_month)
18.          geo_dis = np.zeros((2, 5685, 5567))
19.
20.          files_pre = glob.glob(file_pre + str_i + 'dataset\\????' + str_month + '??' + str_i[0:2] + '.json')
21.          for file in tqdm(files_pre):
22.              with open(file, 'rb') as f:
23.                  dataset = json.load(f)
24.              type_index = np.array(dataset['results']['eddy_type_AE0_CE1']) == 0
25.              a = np.array(dataset['results']['inradius'])[type_index]
26.              b = np.array(dataset['results']['inradius'])[~type_index]
27.              geo_all[0] = np.concatenate([geo_all[0], a])
28.              geo_all[1] = np.concatenate([geo_all[1], b])
29.
30.  fig, (ax, ax2) = plt.subplots(1, 2, figsize=(5, 2), dpi=300)
31.
```

```python
32. plt.subplots_adjust(left=None, bottom=0.19, right=None, top=None, wspace=None, hspace=0.2)
33. ax2.hist(geo_all[1] / 1000 * 2, bins=np.arange(0, 100, 5), color='#000080', label='CE')
34. ax.hist(geo_all[0] / 1000 * 2, bins=np.arange(0, 100, 5), color='#800000', label='AE')
35.
36. ax.grid(ls="--", lw=0.5, color="#4E616C")
37. ax.yaxis.set_major_locator(ticker.MultipleLocator(100 * 3))
38. ax.xaxis.set_major_locator(ticker.MultipleLocator(10))
39. ax.xaxis.set_minor_locator(ticker.MultipleLocator(5))
40. ax.xaxis.set_tick_params(length=2, labelsize=6, which='minor')
41. ax.xaxis.set_tick_params(length=3, labelsize=8, which='major')
42. ax.yaxis.set_tick_params(length=3, labelsize=8)
43. ax.ticklabel_format(style='sci', scilimits=(0, 1), axis='y')
44.
45. ax2.grid(ls="--", lw=0.5, color="#4E616C")
46. ax2.yaxis.set_major_locator(ticker.MultipleLocator(1000 * 3))
47. ax2.xaxis.set_major_locator(ticker.MultipleLocator(10))
48. ax2.xaxis.set_minor_locator(ticker.MultipleLocator(5))
49. ax2.xaxis.set_tick_params(length=2, labelsize=6, which='minor')
50. ax2.xaxis.set_tick_params(length=3, labelsize=8, which='major')
51. ax2.yaxis.set_tick_params(length=3, labelsize=8)
52. ax2.ticklabel_format(style='sci', scilimits=(0, 1), axis='y')
53.
54. fig.text(0.43, 0.03, 'Diameter [km]', fontsize=8)
55. fig.text(0.45, 0.888, '(a)', fontsize=8)
56. fig.text(0.872, 0.888, '(b)', fontsize=8)
57. plt.show()
```

**(13) I strongly recommend the author to polish the language throughout the entire text, as I have identified a significant number of grammar errors and awkward expressions. The Reviewer 1 have provided many language suggestions, but it's not enough to just make changes based on those. Instead, it's advisable to seek assistance from a professional editing service for the revisions.**

Response: Thanks for your comments. I took your advice and polished the whole article.

**L9. 'which obtains from'. Grammatical error.**

Response: Thanks for your comments. I have changed it in line 9. "which is obtained from…"

**L48. 'Compared to the method of SAR images, it can …'. What does 'it' refer to?**

Response: Thanks for your comments. I rephrased it as follows. "Some methods, like SAR and altimeter, only provide physical information about the ocean surface and do not capture biological or chemical processes within the eddies." in lines 62-63.

**L83. Change to 'This is conducted to avoid'.**

**Response:** Thanks for your comments. I have corrected it.

**L228. 'We counted the number of times each grid cell…'. Unclear description.**

**Response:** Thanks for your comments. I rephrased it as follows. "We quantified the frequency of coverage for each grid cell by AEs or CEs and minimized the correlation between...".

**L320. 'ten eight-year periods'. What does this mean?**

**Response:** Thanks for your comments. I rephrased it as follows. "We identified a total of 19,136 anticyclonic eddies and 93,897 cyclonic eddies from eight CHL images per day for ten years..."

**The use of present and past tenses is confusing and inconsistent.**

**I can't point them all out individually. The language does not yet meet the requirements of this journal.**

We feel great thanks for your professional review work on our article and for providing valuable comments. We have diligently revised and polished the entire text to ensure accurate and clear word usage, promoting a better understanding of our research. Thank you again for your review comments, we are more than willing to put in any worthwhile effort to make the article better. If you have any follow-up suggestions or guidance, we would love to hear them as well.

**References**

Choi, J. M. and Kim, W.: Applications of Surface Velocity Current Derived from Geostationary Ocean Color Imager (GOCI), in: 2018 OCEANS-MTS/IEEE Kobe Techno-Oceans (OTO), 1–4, 2018.

Duo, Z., Wang, W., and Wang, H.: Oceanic Mesoscale Eddy Detection Method Based on Deep Learning, Remote Sensing, 11, 1921, https://doi.org/10.3390/rs11161921, 2019.

Franz, K., Roscher, R., Milioto, A., Wenzel, S., and Kusche, J.: Ocean Eddy Identification and Tracking Using Neural Networks, in: IGARSS 2018 - 2018 IEEE International Geoscience and Remote Sensing Symposium, IGARSS 2018 - 2018 IEEE International Geoscience and Remote Sensing Symposium, 6887–6890, https://doi.org/10.1109/IGARSS.2018.8519261, 2018.

Ge, L., Huang, B., Chen, X., and Chen, G.: Medium-Range Trajectory Prediction Network Compliant to Physical Constraint for Oceanic Eddy, IEEE Transactions on Geoscience and Remote Sensing, 61, 1–14, https://doi.org/10.1109/TGRS.2023.3298020, 2023.

Huang, B., Ge, L., Chen, X., and Chen, G.: Vertical Structure-Based Classification of Oceanic Eddy Using 3-D Convolutional Neural Network, IEEE Transactions on Geoscience and Remote Sensing, 60, 1–14, https://doi.org/10.1109/TGRS.2021.3103251, 2022.

Munk, W., Armi, L., Fischer, K., and Zachariasen, F.: Spirals on the sea, Proceedings of the Royal Society of London. Series A: Mathematical, Physical and Engineering Sciences, 456, 1217–1280, https://doi.org/10.1098/rspa.2000.0560, 2000.

Sonali, Sahu, S., Singh, A. K., Ghrera, S. P., and Elhoseny, M.: An approach for de-noising and contrast enhancement of retinal fundus image using CLAHE, Optics & Laser Technology, 110, 87–98, https://doi.org/10.1016/j.optlastec.2018.06.061, 2019.